# EduDial: Constructing a Large-scale Multi-turn Teacher–Student Dialogue Corpus

## Abstract

Recently, several multi-turn dialogue benchmarks have been proposed to evaluate the conversational abilities of large language models (LLMs). As LLMs are increasingly recognized as a key technology for advancing intelligent education, owing to their ability to deeply understand instructional contexts and provide personalized guidance, the construction of dedicated teacher-student dialogue benchmarks has become particularly important. To this end, we present EduDial, a comprehensive multi-turn teacher-student dialogue dataset. EduDial covers 345 core knowledge points and consists of 34,250 dialogue sessions generated through interactions between teacher and student agents. Its design is guided by Bloom's taxonomy of educational objectives and incorporates ten questioning strategies—including situational questioning, zone of proximal development (ZPD) questioning, and metacognitive questioning—thus better capturing authentic classroom interactions. Furthermore, we design differentiated teaching strategies for students at different cognitive levels, thereby providing more targeted teaching guidance. Building on EduDial, we further develop EduDial-LLM 32B via training and propose an 11-dimensional evaluation framework that systematically measures the teaching abilities of LLMs, encompassing both overall teaching quality and content quality. Experiments on 17 mainstream LLMs reveal that most models struggle in student-centered teaching scenarios, whereas our EduDial-LLM achieves significant gains, consistently outperforming all baselines across all metrics.

## 1 Introduction

The burgeoning field of artificial intelligence has drawn significant research focus to the multi-turn conversational capabilities of large language models (LLMs) (Kwon et al., 2024; Duan et al., 2024; Deshpande et al., 2025). As models grow in scale and complexity, a critical challenge lies in evaluating their ability to maintain coherent and meaningful interactions within dynamic dialogue contexts. To systematically address this, a series of multi-turn dialogue benchmarks has been introduced. Noteworthy examples include MT-Bench (Bai et al., 2024), which utilizes powerful LLMs like GPT-4o (Hurst et al., 2024) as evaluators for open-domain conversations, and AlpacaEval (Li et al., 2023), which employs automated metrics to rapidly assess a model's proficiency in multi-turn instruction-following.

However, while these general-purpose benchmarks are useful, they do not fully capture the deeper conversational capabilities of LLMs in vertical domains. As LLM-based agents (Xi et al., 2025; Cai et al., 2025; Shang et al.) and systems play an increasingly important role in education, a competent LLM must not only maintain coherent dialogues but also exhibit core teaching abilities, such as guiding student reasoning, correcting errors, and providing personalized feedback—skills that current general benchmarks fail to assess adequately. Consequently, developing a benchmark that systematically evaluates LLMs' multi-turn conversational abilities in domain-specific contexts, particularly in education, has become an urgent and essential task. As illustrated in Figure 1, (a) represents single-turn teaching, where the model merely delivers knowledge, whereas (b) depicts multi-turn teacher-student dialogue, in which the teacher actively guides the student through interactive instruction.

However, constructing a multi-turn teacher-student dialogue benchmark still faces two major challenges: **(1) How to accurately determine the optimal timing for teacher questioning?** Inappropriate timing can interrupt students' comprehension, disrupt their reasoning process, and ultimately

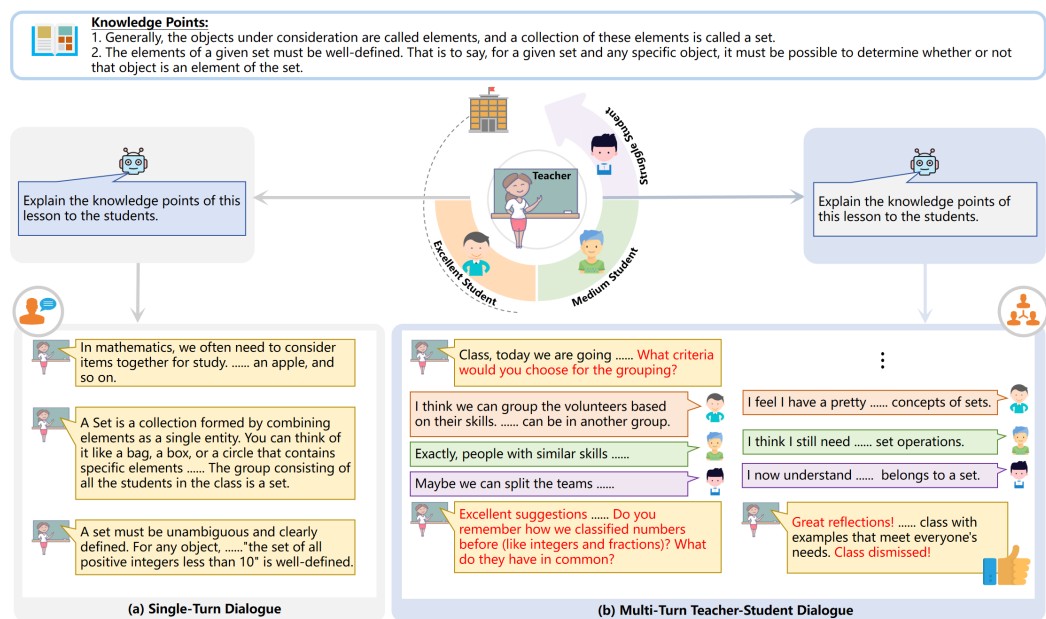

Figure 1: Comparison of "(a) single-turn " and "(b) mutil-turn teacher-student" dialogue.

hinder learning outcomes. For instance, when students are still grasping a new concept, frequent questioning may interfere with their cognitive flow, significantly reducing learning efficiency. **(2) How to adjust questioning strategies according to different teaching stages?** Existing approaches often lack stage-specific strategies. Even with proper timing, unsuitable strategies may backfire and negatively impact learning. For example, posing questions beyond a student's cognitive level at the initial stage of learning can easily lead to frustration and hinder further progress.

To address these challenges, we propose **EduDial**, a multi-turn teacher-student dialogue dataset designed to enable LLMs to pose appropriate questions at the right time in education. EduDial covers 345 core knowledge points and comprises 34,250 dialogue sessions generated through interactions between teacher and student. The dataset follows five progressive teaching stages: introduction, concept exploration, deep understanding, knowledge application and reflection, and incorporates ten questioning strategies—including situational, zone of proximal development (ZPD), and metacognitive questioning—to better simulate authentic classroom interactions. Moreover, differentiated teaching strategies are designed for students at varying cognitive levels, providing more targeted guidance. Building on EduDial, we further train **EduDial-LLM 32B** and propose an 11-dimensional evaluation framework that systematically measures LLMs' teaching capabilities, encompassing both overall teaching quality and content quality. Extensive experiments demonstrate the challenging nature of EduDial and the effectiveness of our approach in enhancing LLMs' performance in educational tasks.

## 2 RELATED WORK

**Dialogue Dataset.** Early dialogue data research primarily focuses on single-turn interaction scenarios, constructing domain-specific question-answering datasets that enhance model performance in healthcare, finance, and code generation through supervised fine-tuning (Zeng et al., 2020; Chen et al., 2021; Dong et al., 2024). However, single-turn dialogues cannot capture crucial elements such as dialogue history understanding and adaptive response strategies, which severely restricts the practical deployment of LLMs. Recognizing these limitations, researchers shift toward multi-turn dialogue systems. Current research explores both general and domain-specific approaches. General methods establish evaluation benchmarks through iterative optimization but lack domain adaptability (Wu et al., 2025; Bai et al., 2024). Domain-specific methods construct specialized datasets in law, medicine, and mathematics through role simulation (ShengbinYue et al., 2025; Wang et al., 2024a; Liu et al., 2025c). However, these datasets primarily focus on knowledge transfer accuracy and fail to meet educational requirements that guide student thinking through interaction. Therefore, researchers increasingly focus on multi-turn educational dialogue datasets. Early research relies on costly human-annotated

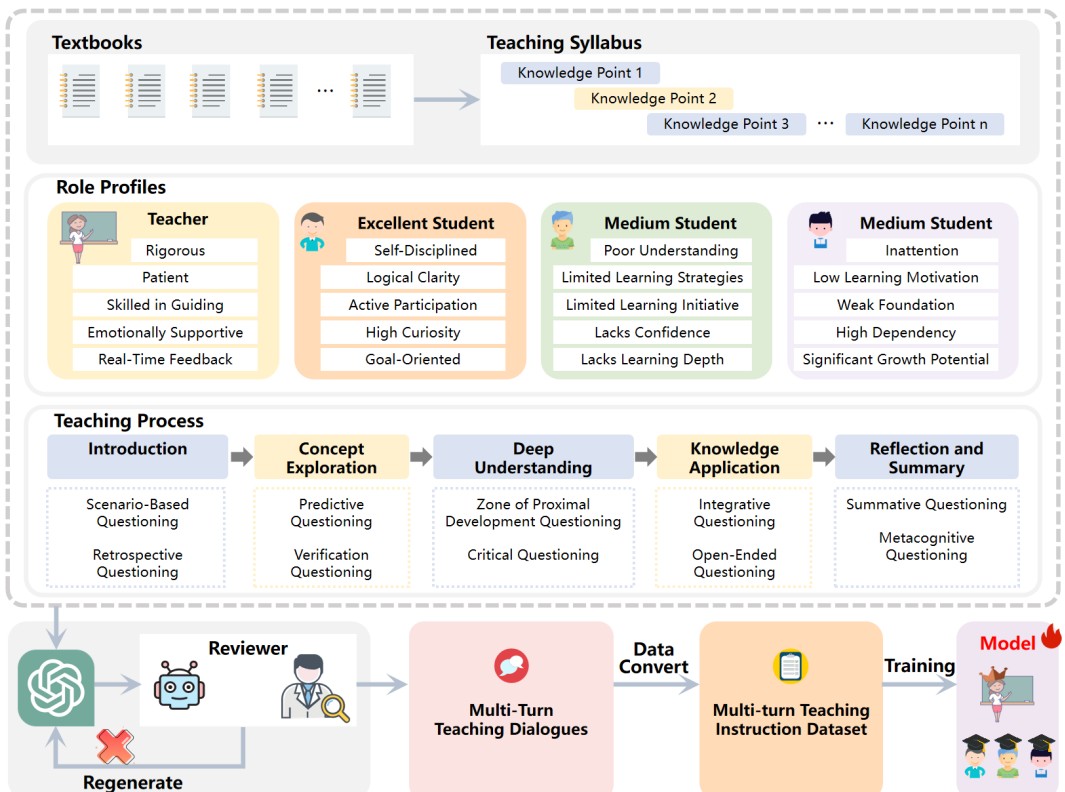

Figure 2: Based on the teaching syllabus and role profiles, generate high-quality multi-turn teaching instruction data through a five-stage teaching process to train one teacher and three student models.

data (Stasaski et al., 2020; Caines et al., 2020; Suresh et al., 2022). With LLM development, recent studies employ various approaches, including textbook-based generation (Wang et al., 2024b), pedagogical theory integration (Liu et al., 2025b), and Socratic teaching simulation through teacher-LLM collaboration (Macina et al., 2023) or multi-agent frameworks (Liu et al., 2024b). However, existing educational dialogue datasets employ rigid questioning strategies and ignore cognitive differences among students. To address these limitations, we propose a five-stage instructional framework that defines when and how to ask questions at each teaching stage, along with differentiated strategies tailored to varying cognitive levels.

**Leveraging LLMs for Advanced Intelligent Education.** Large language models (LLMs) profoundly reshape three key application directions of intelligent education: the automation of teaching content generation, learning assessment and feedback, and interactive teaching (Labadze et al., 2023; Stamper et al., 2024; Wang et al., 2024c). For teaching content, current research focuses on using LLMs to automatically generate teaching plans, exercises, and PPTs based on specific goals (Hu et al., 2024; Li et al., 2024; Xie et al., 2025). It provides personalized and diverse teaching support for teachers. For learning assessment and feedback, LLMs demonstrate the potential to evaluate learners' cognitive levels and provide personalized feedback instantly (Meyer et al., 2024). It helps create more responsive and adaptive educational experiences for students. Regarding interactive teaching, most relevant to our research, there are two main research directions. The first direction uses prompt engineering (Liu et al., 2023) to directly leverage general large language models for teaching guidance. Examples include providing support in programming (Kargupta et al., 2024), classroom education (Zhang et al., 2024), and psychological counseling (Qiu and Lan, 2024). The second direction enhances the model's multi-turn interactive teaching ability through specialized training and fine-tuning. For example, previous work enhances deep interactive teaching abilities by collecting numerous teaching instructions to train models (like SocraticLM (Liu et al., 2025a)).

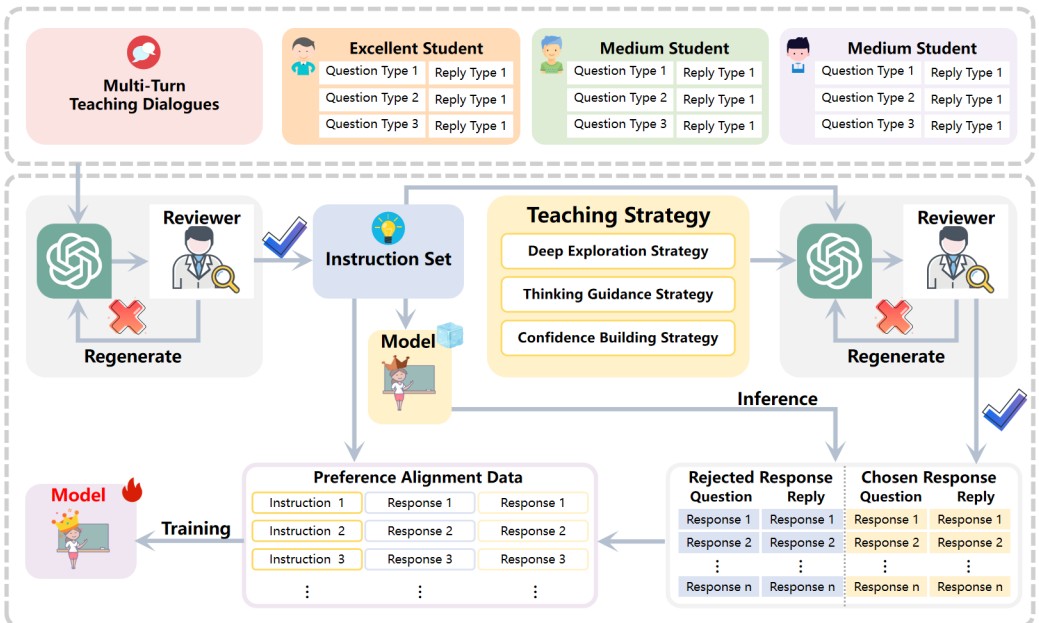

Figure 3: Develop corresponding teaching strategies based on the three student response types, construct preference datasets by generating chosen and rejected responses, and further optimize the teacher model using direct preference optimization.

# 3  THE EDUDIAL DATASET

## 3.1  TASK DEFINITION

**Single-turn dialogue** represents the most basic form of interaction. Each sample contains a standalone independent question-answer pair without contextual dependencies. This data type evaluates models' immediate direct response capabilities to individual queries. It encompasses factual question answering, task execution, and information retrieval scenarios.

**Multi-turn dialogue** extends interactions to continuous conversational sequences. In this paradigm, models must maintain dialogue history and understand topic evolution. They generate coherent responses based on prior interactions. Multi-turn dialogue embodies natural conversational dynamics through progressive topic refinement, incremental knowledge construction, and coherent reasoning progression. It requires models to possess context understanding and dialogue state tracking.

**Teacher-student multi-turn dialogue** introduces role asymmetry and objective divergence to multi-turn interactions. Unlike general multi-turn dialogues, teacher-student conversations exhibit clearly defined role positioning. Teachers guide and explain; students express understanding and confusion. This interaction paradigm necessitates dynamic pedagogical adaptation based on student feedback signals. Teachers identify comprehension levels, calibrate explanatory approaches, and deliver tailored scaffolding. Simultaneously, students demonstrate different cognitive states from bewilderment to partial grasp to full application proficiency.

## 3.2  PREPARATION

Unlike general-domain data synthesis, educational dialogue generation relies on pedagogical theories, teaching strategies, and domain knowledge. We therefore establish three foundational components during the preparation.

*1) Teaching Syllabus.* To ensure the authenticity of teaching content and comprehensive coverage of core knowledge points, we surveyed over 100 primary and secondary schools. Additionally, we held extensive discussions with experienced mathematics teachers. Based on this, we selected 345 core knowledge points most concerned about teachers and students from the K-12 mathematics curriculum,

covering 173 chapters. Furthermore, we extracted each knowledge point from textbooks, including concepts, theorems, and examples, to ensure consistency with curriculum standards.

*2) Teaching Process and Question Strategies.* We propose a teaching process that integrates Bloom's taxonomy (Forehand, 2010), constructing a five-stage progressive teaching process where each stage aligns with specific cognitive levels. The introduction stage corresponds to "remember" and activates prior knowledge through contextual and retrospective questioning. The concept exploration stage maps to "understand" and facilitates concept internalization via predictive and verification questioning. The deep understanding stage aligns with "analyze" to cultivate critical thinking using the zone of proximal development and critical questioning. The knowledge application stage combines "apply and evaluate" to enhance practical competencies through integrative and open-ended questioning. Finally, the reflection and summary stage corresponds to "create" and promotes knowledge reconstruction with summary and metacognitive questioning. These question strategies derive from extensive analysis of MOOC instructional videos and in-depth discussions with experienced mathematics teachers, ensuring each teaching phase has appropriate dual question strategies.

*3) Role Profiles and Teaching Strategies.* We establish a four-member role system comprising a teacher and three student profiles (excellent, medium, struggling), characterized through five dimensions: cognitive capacity, learning motivation, classroom engagement, learning strategies, and self-regulation skills. Based on these profiles, we design differentiated teaching strategies. Excellent students exhibit inquiry-based, exploratory, and optimization-focused questioning patterns, with responses demonstrating accuracy, clarity of explanation, and the ability to connect related concepts. For these students, we employ a deep exploration strategy that fosters higher-order thinking through progressive questioning sequences. Medium students typically pose application-oriented, procedural, and detail-focused questions, with responses showing partial correctness, uncertainty expression, and hint-seeking features. Accordingly, we implement a thinking guidance strategy to support their problem-solving process and enhance cognitive clarity. Struggling students mainly ask fundamental, understanding, and learning efficacy questions, with answers often showing errors, vagueness, or absence. For this group, we develop a confidence-building strategy that strengthens foundational cognition and learning confidence through gradual knowledge construction.

## 3.3 DATASET CONSTRUCTION

**Data Generation** We detail the construction process of our dataset, EduDial, which is composed of two main components: Multi-turn Teaching Instruction (MTI) and the Preference Dataset based on Teaching Strategies (PDTS).

*MTI Dataset Generation.* We design a three-step data generation pipeline for multi-turn teacher-student dialogues in the MTI dataset. Inspired by MT-Bench, we begin by transforming the teaching process, role profiles, and teaching syllabus into structured prompt templates. Using LLMs (i.e., o1 (Jaech et al., 2024)), we generate multi-turn teaching dialogues that maintain strict role consistency and logical coherence, ensuring interactions align with authentic teaching scenarios. In the second step, we collaborate with experts to establish five core teaching principles: teaching stage integrity, comprehensive questioning strategy, consistent role-playing, accurate knowledge delivery, and natural interaction flow. Based on these principles, we ensure that the generated dialogues align with teaching standards through an expert-machine dual validation. Finally, in step three, we convert dialogues that meet the established standards into role-specific supervised fine-tuning (SFT) corpora. For the teacher model, the training data pairs student responses as input with teacher responses as output. For the student model, the training data uses teacher and peer responses as input and the student responses as output.

*PDTS Dataset Generation.* The PDTS Dataset Generation process is based on differentiated teaching strategies and involves a systematic data generation approach. First, we combine student profile descriptions at each cognitive level with multi-turn teaching dialogues to create structured prompts. These prompts guide LLMs to generate student interactions that align with specific cognitive levels, with the generated questions and replies serving as "instructions" for the teacher model input. In the second step, we input these instructions into a fine-tuned teacher model to generate rejected responses. While these responses are reasonable in content, they deviate from the predetermined teaching strategies. Next, in step three, we combine the instructions with the corresponding teaching strategies and input them into LLMs to generate chosen responses, which accurately reflect the

designated teaching methods. Finally, in step four, we construct preference alignment triplets containing ⟨instruction, chosen response, rejected response⟩, which form the final PDTS dataset for direct preference optimization (DPO) training.

**Expert-Machine Dual Verification.** We employ a dual verification system combining expert evaluation and machine-based assessment to ensure data quality. All generated data undergo automatic verification by GPT-4o and manual review by experts. For the MTI dataset, verification criteria include: logical progression of teaching stages, consistency between questioning strategies and objectives, alignment between student responses and cognitive levels, the educational value of teacher feedback, and the natural flow of dialogue. For the PDTS dataset, verification dimensions encompass: teaching strategy adaptability and distinction between chosen and rejected responses. For data failing verification, expert annotators and GPT-4o provide feedback for regeneration. This process continues until the data meet established quality standards.

### 3.4 Two-stage Training Strategy

One of our main contributions is constructing two datasets. To validate their effectiveness, we adopt a standard two-stage training framework: SFT followed by DPO. We use these mainstream methods to ensure that performance gains stem from data quality rather than algorithmic innovations.

**SFT Stage.** To ensure that LLMs better align with role profiles when simulating different roles, we use instruction fine-tuning techniques to train four LLMs. Specifically, we use instruction fine-tuning data for four different roles and train four LLMs using the negative log-likelihood loss function. The loss function is defined as follows:

$$\mathcal{L}_{sft} = -\sum_{c=1}^{C} \sum_{k=1}^{K_c} \sum_{i=1}^{N_{c,k}} \log P\left(t_{c,k,i} \mid t_{c,k,<i}, \ C_{c,k}\right). \tag{1}$$

$$C_{c,k} = \left\{ \left(input_{c,j}, output_{c,j}\right) \right\}_{j=1}^{k-1} \cup \{input_{c,k}\}, \tag{2}$$

where $C$ is the batch size, $K_c$ is the total number of rounds for the c-th sample, $N_{c,k}$ is the number of tokens output in the k-th round of the c-th sample, $t_{c,k,i}$ is i-th token output in the k-th round of the c-th sample, $t_{c,k,<i}$ is the sequence of tokens output before i-th token in the k-th round of the c-th sample, $C_{c,k}$ is the context before the k-th round of the c-th sample, $input_{c,j}$ is the input of the j-th round in the c-th sample, and $output_{c,j}$ is the output of the j-th round in the c-th sample.

**DPO Stage.** To further enhance the model's teaching capabilities, we train the fine-tuned teacher model using the direct preference alignment algorithm. The algorithm optimizes two objectives: maximizing the generation probability of chosen responses and minimizing the generation probability of rejected responses. The optimization mechanism ensures that the model tends to generate responses conforming to teaching strategies while suppressing responses deviating from teaching strategies. The loss function of the direct preference alignment algorithm is defined as follows.

$$\mathcal{L}_{dpo} = -\log \sigma(s_{chosen} - s_{rejected}), \tag{3}$$

$$s_{chosen} = \log P(chosen \mid instruction), \tag{4}$$

$$s_{rejected} = \log P(rejected \mid instruction), \tag{5}$$

where $s_{chosen}$ and $s_{rejected}$ denote the scores of the chosen and rejected responses, respectively, and $\sigma$ represents the sigmoid function.

### 3.5 Evaluation Metrics.

To evaluate teaching quality, we design two metric categories: overall and content quality. Overall quality includes nine dimensions: insight, response, feedback, thinking, fluency, interactivity, emotional support, adaptability, and goal. Content quality covers relevance (syllabus alignment) and coverage (knowledge point inclusion). Table 1 details all eleven dimensions. Overall quality and relevance are rated on a five-point Likert scale (1-5), while coverage is measured as a percentage, with 100Evaluation combines human and machine methods: five experts score independently, and their average forms the human result. GPT-4o independently evaluates the same data five times, with the average as the machine result.

Table 1: Evaluation Metrics for Overall and Content Quality.

| Dimension | Abbr. | Definition |
|---|---|---|
| *Overall Quality* | | |
| Insight | INS | Assess whether the teacher accurately identifies student learning needs, knowledge levels, and question intent. |
| Response | RES | Assess whether the teacher effectively addresses student questions and provides feasible, constructive guidance. |
| Feedback | FB | Assess whether the teacher provides timely and effective feedback that facilitates improvement. |
| Thinking | THK | Assess whether the teacher fosters thinking skills, including analysis and open-mindedness. |
| Fluency | FLU | Assess whether the teacher communicates clearly and is easy to understand. |
| Interactivity | INT | Assess the frequency and quality of interactions between teacher and students in teaching dialogues. |
| Emotional Support | EMO | Assess whether the teacher provides emotional support during teaching and creates a positive learning environment. |
| Adaptability | ADP | Assess whether the teacher adjusts teaching methods to accommodate different contexts, ensuring personalized guidance. |
| Goal | GOL | Assess whether teacher guidance helps achieve teaching goals, such as knowledge acquisition. |
| *Content Quality* | | |
| Relevance | REL | Assess whether the teacher-student dialogue is relevant to the teaching content of this chapter. |
| Coverage | COV | Assess the proportion of required teaching knowledge points encompassed within the teacher-student dialogue. |

## 3.6 STATISTICS

We split the dataset at an 8:2 ratio by chapter. The training set contains 137 chapters (275 knowledge points), and the test set contains 36 chapters (70 knowledge points). We construct two datasets: (1) MTI Dataset: We generate 100 teacher-student dialogues per training chapter, yielding 13,700 dialogue samples. Each dialogue averages 11 interaction rounds. Teachers pose an average of 14.2 questions per dialogue. (2) PDTS Dataset: We generate 50 instructions per student role. Each instruction yields one preference pair, totaling 20,550 preference pairs. For evaluation, we create 36 teaching scenarios per model. Each model acts as the teacher, engaging with three students of varying cognitive levels. Sessions end when the teacher completes instruction or after 15 rounds.

## 4 EXPERIMENTS

### 4.1 EXPERIMENTAL SETUP

**Baselines.** We select 18 large language models as baselines, including nine closed-source and eight open-source models. For closed-source models, we evaluate Gemini-2.5-Pro-Exp (Google DeepMind, 2025), Gemini-2.0-Flash-Exp (Google Cloud, 2025), Claude-3-7-Sonnet (Anthropic, 2025), Claude-3-7-Sonnet-Thinking (Anthropic, 2025), Claude-3-5-Sonnet (Anthropic, 2024), o3-mini-High (OpenAI, 2025), o1 (Jaech et al., 2024), GPT-4o (Hurst et al., 2024) and GPT-3.5 (Ouyang et al., 2022). Regarding open-source models, we evaluate DeepSeek-R1 (Guo et al., 2025), DeepSeek-V3 (Liu et al., 2024a), Llama-3.3-70B-Instruct (Meta AI, 2024), Yi-1.5-34B-Chat (Young et al., 2024), Qwen-2.5-72B-Instruct (Yang et al., 2024a), Qwen-2.5-72B-Math (Yang et al., 2024b), QwQ-32B-Preview (Qwen Team, 2024), SocraticLM (Liu et al., 2024b), and ours model EduDial-LLM.

**Experiments Settings.** We conduct experiments on two NVIDIA A100 80GB GPUs with CUDA 12.4, PyTorch 2.4.0, and Python 3.10. In the SFT stage, we use the QLoRA (Dettmers et al., 2023) with 4-bit quantization to fine-tune models, and LoRA (Hu et al.) configuration (rank r=64, scaling factor $\alpha = 16$). We utilize the AdamW optimizer alongside BF16 mixed precision training while evaluating model performance every 500 steps. Each epoch in this stage takes approximately 3 hours to complete, with a total computation time of 45 hours across all four models. For more experimental settings, see Table 5 in the appendix. In the DPO stage, the learning rate is 5e-7 with a batch size of 8. We accumulate gradients over 16 steps and train for two epochs. The QLoRA (r=64, $\alpha$=16) configuration remains consistent with the first stage. For DPO-specific parameters, we set $\beta$ to 0.1, and the maximum sequence length is 1024 tokens. Each epoch requires approximately 4 hours, resulting in a total computation time of 8 hours for this stage. All hyperparameters are empirically determined via grid search on the test set.

### 4.2 PERFORMANCE COMPARISON

Our model exhibits comprehensive and balanced performance across nine evaluation dimensions of overall quality, achieving the highest average scores in both machine-based and human evaluations. In contrast, baseline models show uneven performance across different dimensions. Furthermore,

Table 2: Comprehensive Machine (M) and Human (H) Evaluation of 18 Models across Overall Quality and Content Quality.

| Model | Overall Quality | | | | | | | | | | | | | | | | | | | | | Content Quality | | | |
|---|---|---|---|---|---|---|---|---|---|---|---|---|---|---|---|---|---|---|---|---|---|---|---|---|---|
| | INS | | RES | | FB | | THK | | INT | | EMO | | ADP | | FLU | | GOL | | AVG | | REL | | COV(%) | |
| | M | H | M | H | M | H | M | H | M | H | M | H | M | H | M | H | M | H | M | H | M | H | M | H |
| **Closed Source Models** | | | | | | | | | | | | | | | | | | | | | | | | |
| Gemini-2.5-Pro-Exp (Google DeepMind, 2025) | 4.06 | 4.05 | 4.23 | 4.41 | 3.84 | 3.95 | 3.48 | 3.71 | 4.56 | 4.58 | **4.67** | **4.64** | 4.12 | 3.97 | **4.67** | **4.62** | 4.09 | 4.08 | 4.19 | 4.22 | **4.95** | **4.81** | 94.17 | 91.98 |
| Gemini-2.0-Flash-Exp (Google Cloud, 2025) | 3.89 | 3.78 | 4.34 | 4.37 | 3.89 | 4.00 | 4.29 | 4.41 | 3.57 | 3.51 | 3.69 | 4.01 | 3.89 | 3.79 | 4.46 | 4.64 | 4.31 | 4.29 | 4.04 | 4.09 | 4.94 | 4.78 | **96.16** | **94.84** |
| Claude-3-7-Sonnet (Anthropic, 2025) | 3.90 | 3.93 | 4.33 | 4.29 | 3.67 | 3.84 | 3.39 | 3.51 | 4.36 | 4.37 | 4.39 | 4.32 | 3.91 | 3.93 | 4.50 | 4.59 | 4.01 | 3.98 | 4.05 | 4.08 | 4.89 | 4.76 | 90.08 | 88.47 |
| Claude-3-7-Sonnet-Thinking (Anthropic, 2025) | 4.05 | 3.91 | **4.63** | **4.50** | 3.85 | 3.71 | 4.19 | 4.08 | 3.42 | 3.11 | 3.28 | 3.34 | 3.63 | 3.53 | 4.63 | 4.61 | 4.29 | 4.29 | 4.00 | 3.90 | 4.60 | 4.49 | 90.26 | 89.31 |
| Claude-3-5-Sonnet (Anthropic, 2024) | 3.82 | 3.71 | 4.29 | 4.00 | 3.68 | 3.40 | 3.44 | 3.38 | 3.62 | 3.21 | 3.97 | 3.86 | 3.79 | 3.49 | 4.47 | 4.14 | 3.94 | 4.05 | 3.89 | 3.69 | 4.80 | 4.68 | 93.70 | 91.82 |
| o3-mini-High (OpenAI, 2025) | 3.66 | 3.45 | 4.24 | 4.10 | 3.41 | 3.18 | 3.23 | 3.30 | 2.86 | 2.60 | 3.40 | 3.23 | 2.91 | 2.78 | 4.42 | 4.30 | 3.76 | 3.77 | 3.54 | 3.41 | 4.20 | 4.12 | 80.11 | 79.24 |
| o1 (Jaech et al., 2024) | 3.50 | 3.36 | 4.30 | 4.30 | 3.20 | 3.17 | 3.42 | 3.51 | 2.66 | 2.62 | 3.04 | 2.86 | 2.82 | 2.71 | 4.40 | 4.34 | 3.91 | 3.47 | 3.47 | 3.42 | 4.54 | 4.46 | 90.17 | 92.91 |
| GPT-4o (Hurst et al., 2024) | 3.75 | 3.63 | 4.14 | 4.08 | 3.61 | 4.04 | 3.92 | 3.92 | 3.28 | 3.00 | 3.47 | 2.80 | 3.47 | 2.86 | 4.31 | 4.27 | 3.81 | 4.20 | 3.75 | 3.64 | 4.83 | 4.73 | 88.52 | 84.89 |
| GPT-3.5 (Ouyang et al., 2022) | 3.67 | 3.50 | 4.03 | 3.85 | 3.22 | 3.10 | 3.92 | 3.84 | 2.92 | 2.58 | 3.25 | 2.56 | 3.28 | 2.94 | 4.31 | 3.93 | 3.72 | 3.86 | 3.59 | 3.35 | 4.76 | 4.61 | 91.30 | 89.83 |
| **Open Source Models** | | | | | | | | | | | | | | | | | | | | | | | | |
| Deepseek-R1 (Guo et al., 2025) | 3.77 | 3.54 | 4.50 | 4.42 | 3.61 | 3.44 | 3.79 | 4.03 | 2.73 | 2.62 | 2.76 | 2.58 | 3.40 | 3.27 | 4.18 | 4.23 | 4.16 | 4.17 | 3.66 | 3.59 | 4.57 | 4.48 | 90.18 | 89.93 |
| Deepseek-V3 (Liu et al., 2024a) | 3.39 | 3.25 | 4.09 | 4.26 | 3.21 | 3.09 | 3.08 | 3.25 | 2.50 | 2.31 | 2.72 | 2.58 | 2.77 | 2.74 | 4.07 | 4.35 | 3.94 | 3.91 | 3.31 | 3.30 | 4.62 | 4.51 | 95.04 | 94.32 |
| Llama-3.3-70B-Instruct (Meta AI, 2024) | 2.56 | 2.04 | 3.08 | 1.97 | 2.53 | 2.50 | 2.36 | 2.25 | 2.81 | 2.15 | 2.72 | 2.23 | 2.47 | 2.06 | 4.25 | 3.47 | 3.56 | 3.36 | 2.93 | 2.45 | 4.33 | 4.12 | 88.19 | 87.69 |
| Yi-1.5-34B-Chat (Young et al., 2024) | 2.28 | 2.10 | 3.11 | 3.17 | 2.81 | 2.86 | 2.28 | 2.28 | 2.75 | 2.92 | 2.89 | 3.08 | 2.44 | 2.78 | 3.64 | 3.22 | 3.58 | 3.81 | 2.86 | 2.91 | 4.28 | 4.11 | 86.92 | 85.53 |
| Qwen-2.5-72B-Instruct (Yang et al., 2024a) | 3.18 | 2.89 | 3.45 | 2.74 | 2.91 | 2.43 | 2.70 | 2.89 | 2.59 | 2.78 | 2.97 | 2.67 | 2.24 | 2.41 | 3.30 | 3.62 | 2.97 | 3.39 | 2.92 | 2.87 | 4.48 | 4.27 | 89.42 | 87.75 |
| Qwen-2.5-72B-Math (Yang et al., 2024b) | 2.02 | 3.06 | 2.32 | 3.60 | 1.96 | 2.63 | 1.93 | 2.58 | 1.91 | 2.57 | 1.89 | 2.44 | 1.84 | 2.01 | 2.84 | 3.26 | 2.70 | 3.19 | 2.16 | 2.82 | 4.35 | 4.16 | 83.89 | 82.43 |
| QwQ-32B-Preview (Qwen Team, 2024) | 2.89 | 2.22 | 3.24 | 2.42 | 2.65 | 2.28 | 2.43 | 2.33 | 2.69 | 2.25 | 2.67 | 2.31 | 2.16 | 2.22 | 3.15 | 3.44 | 2.89 | 3.31 | 2.75 | 2.53 | 4.43 | 4.25 | 88.08 | 86.56 |
| SocraticLM (Liu et al., 2024b) | 2.69 | 2.97 | 3.69 | 3.96 | 3.31 | 3.48 | 3.08 | 3.07 | 3.56 | 3.46 | 3.56 | 3.43 | 2.83 | 2.82 | 3.58 | 3.94 | 4.03 | 3.81 | 3.37 | 3.44 | 4.39 | 4.31 | 87.94 | 86.31 |
| EduDial-LLM (Ours) | **4.58** | **4.43** | 4.12 | 4.26 | **4.62** | **4.25** | **4.55** | **4.41** | **4.64** | **4.68** | 4.10 | 4.09 | **4.60** | **4.25** | 4.43 | 4.49 | **4.56** | **4.42** | **4.47** | **4.36** | 4.83 | 4.77 | 91.83 | 88.29 |

most open-source models score below 4.0 in multiple dimensions, highlighting their limitations. Regarding content quality, our scores match the best baseline models. This can be attributed to our carefully constructed high-quality dataset, EduDial. The MTI dataset used for SFT stage training endows our model with clear teaching objectives and reliable content processing capabilities. These capabilities manifest in outstanding performance across goal, relevance, and coverage dimensions. Additionally, our model masters questioning techniques that effectively promote teacher-student interaction and stimulate student thinking. These skills lead to excellent performance in dimensions of thinking and insight. Subsequently, the PDTS dataset used for DPO stage training significantly enhances the model's ability to perceive varied student states. The model learns to make real-time adjustments based on student responses. Superior performance in interactivity, adaptability, and feedback dimensions substantiates this improvement. In conclusion, our dataset, EduDial, combined with the two-stage training paradigm, enables the model to maintain deep cognitive interaction while achieving teaching objectives.

## 4.3 ABLATION STUDY

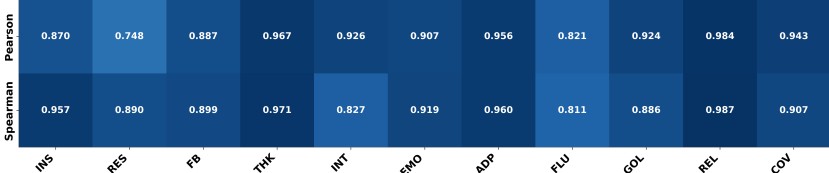

Figure 4: Machine-Human Rating Consistency: Pearson and Spearman Correlation Coefficients Between Machine and Human Evaluations for 18 Models Across 11 Dimensions.

**Human-Machine Rating Consistency Analysis.** As shown in Fig. 4, we evaluated the consistency between human and machine ratings across 11 dimensions using Pearson and Spearman correlation coefficients. The results demonstrate robust inter-rater consistency with average Pearson and Spearman coefficients of 0.90 and 0.91, respectively. Analysis of all dimensions shows that content quality aspects (Relevance and Coverage) consistently achieve high correlation coefficients exceeding 0.90. In contrast, the nine overall quality dimensions exhibit variable performance. While thinking and adaptability dimensions demonstrate high consistency, interactivity, feedback, and insight show comparatively lower agreement. While machine evaluation systems align strongly with human expert assessment for dimensions with explicit criteria and objective metrics, they show decreased agreement on dimensions requiring complex contextual understanding and teaching interaction quality

assessment. Despite these variations, the overall high consistency indicates machine rating reliability. Therefore, we employ machine ratings as the primary assessment method in subsequent experiments.

**Impact of Training Strategies on Teaching Capabilities.** Table 3 shows evaluation results across 11 dimensions for the complete model and three ablation models when teaching students of different performance levels. The experimental results demonstrate: In overall quality, the complete model achieves optimal performance across all student types. It significantly outperforms other ablation models. Among single-stage training approaches, SFT significantly enhances base model performance, effectively teaching the model questioning techniques. Similarly, DPO notably strengthens thinking and adaptability dimensions while yielding modest interactivity gains, indicating its efficacy for complex teaching capabilities. Moreover, two-stage training optimizes performance beyond SFT alone, primarily bolstering thinking and adaptability dimensions. Analysis across different performance levels shows that the complete model possesses excellent adaptive capabilities. It provides personalized teaching according to different learning needs. Aspects of content quality assessment exhibit minimal fluctuation across all models. However, the complete model still achieves notable improvements. Experimental results prove that two-stage training ensures accuracy and comprehensiveness in knowledge transfer during teaching.

Table 3: EduDial-LLM One-on-One Teaching Performance Across Different Training Configurations and Student Proficiency Levels. Student Types: Excellent (E), Medium (M), and Struggling (S).

| Model | Overall Quality | | | | | | | | | | Content Quality | |
|---|---|---|---|---|---|---|---|---|---|---|---|---|
| | INS | RES | FB | THK | INT | EMO | ADP | FLU | GOL | AVG | REL | COV(%) |
| **Complete Model (QwQ-32B-Preview + SFT + DPO)** | | | | | | | | | | | | |
| EduDial-LLM (E) | 4.28 | 4.49 | 4.61 | 4.27 | 4.31 | 4.12 | 4.78 | 4.70 | 4.86 | 4.49 | 4.78 | 91.83 |
| EduDial-LLM (M) | 4.23 | 4.41 | 4.52 | 4.24 | 4.25 | 4.19 | 4.64 | 4.89 | 4.75 | 4.46 | 4.68 | 90.87 |
| EduDial-LLM (S) | 4.27 | 4.46 | 4.59 | 4.25 | 4.42 | 4.18 | 4.58 | 4.77 | 4.83 | 4.48 | 4.74 | 91.42 |
| **SFT Only (QwQ-32B-Preview + SFT)** | | | | | | | | | | | | |
| EduDial-LLM w/o DPO (E) | 4.16 | 4.40 | 4.31 | 3.92 | 4.18 | 3.94 | 4.17 | 4.83 | 4.53 | 4.27 | 4.63 | 89.03 |
| EduDial-LLM w/o DPO (M) | 4.14 | 4.43 | 4.38 | 3.81 | 4.19 | 3.98 | 4.14 | 4.79 | 4.47 | 4.26 | 4.64 | 88.31 |
| EduDial-LLM w/o DPO (S) | 4.08 | 4.45 | 4.32 | 3.89 | 4.14 | 3.91 | 4.13 | 4.77 | 4.45 | 4.24 | 4.66 | 89.14 |
| **DPO Only (QwQ-32B-Preview + DPO)** | | | | | | | | | | | | |
| EduDial-LLM w/o SFT (E) | 3.21 | 3.83 | 3.95 | 3.99 | 3.96 | 3.65 | 3.72 | 4.02 | 3.98 | 3.81 | 4.45 | 87.25 |
| EduDial-LLM w/o SFT (M) | 3.15 | 3.72 | 3.56 | 3.93 | 3.82 | 3.51 | 3.64 | 3.85 | 4.12 | 3.70 | 4.38 | 86.34 |
| EduDial-LLM w/o SFT (S) | 3.29 | 3.89 | 3.71 | 3.87 | 3.91 | 3.69 | 3.75 | 3.93 | 3.78 | 3.76 | 4.43 | 89.05 |
| **Base Model (QwQ-32B-Preview)** | | | | | | | | | | | | |
| EduDial-LLM w/o SFT w/o DPO (E) | 2.67 | 3.11 | 3.33 | 2.56 | 3.33 | 3.31 | 2.33 | 3.94 | 3.33 | 3.10 | 4.39 | 89.83 |
| EduDial-LLM w/o SFT w/o DPO (M) | 2.28 | 3.06 | 2.44 | 2.17 | 3.00 | 2.86 | 2.86 | 3.44 | 3.78 | 2.88 | 4.29 | 82.47 |
| EduDial-LLM w/o SFT w/o DPO (S) | 2.83 | 3.17 | 2.97 | 2.14 | 3.14 | 3.22 | 2.61 | 3.75 | 3.06 | 2.99 | 4.37 | 88.58 |

**Balancing Teaching Capability and Mathematical Reasoning.** To verify our model's general reasoning capability, we conduct experiments on two challenging mathematical reasoning datasets: Math500 (Hendrycks et al.) and AIME2024. Results show in Table 4 our model achieves 87.54% Averagepass@1 on Math-500 (Baseline: 91.12%) and 33.33% on AIME2024 (Baseline: 43.33%). Although our model performs slightly below the baseline in mathematical reasoning, this difference falls within the expected range. Our model aims not at maximizing reasoning performance but at enhancing teaching capability. In educational contexts, both reasoning capabilities and teaching skills are indispensable components. An ideal educational assistant provides accurate answers while identifying learners' confusion points and offering personalized guidance. As our model exemplifies, it maintains acceptable mathematical reasoning abilities while placing greater emphasis on guided teaching capabilities. These features make our model more suitable for real-world educational applications.

Table 4: Reasoning Capability Comparison Between Base Model and EduDial-LLM on Standard Benchmarks.

| Model | Math500 | AIME2024 |
|---|---|---|
| QwQ-32B-Preview | **91.12** | **43.33** |
| EduDial-LLM | 87.54 | 33.33 |

## 5 CONCLUSION

In this work, we introduce EduDial-LLM, a novel large language model tailored for multi-turn teacher-student dialogue teaching. Departing from the traditional single-turn paradigm, EduDial-LLM actively engages students through a structured questioning process, emulating effective pedagogical practices found in real-world classrooms. Central to our approach is the EduDial dataset, which supports two-stage training through instructional fine-tuning and preference optimization, enabling the model to adapt its teaching strategies to diverse learner profiles. To rigorously evaluate teaching quality, we also propose a comprehensive 11-dimensional evaluation framework. Experimental results on 17 mainstream LLMs validate the effectiveness of our approach: while most models struggle with student-driven interaction, EduDial-LLM consistently achieves superior performance across all evaluation dimensions. This work underscores the value of guided, interactive teaching in intelligent education and offers a new direction for future LLM-based tutoring systems.

**Ethics statement.** The EduDial dataset was constructed using simulated interactions between teacher and student agents. No real-world student data or personally identifiable information was used in the creation of this dataset, thereby eliminating privacy concerns. Furthermore, the content was carefully curated and validated to ensure the integrity, accuracy, and safety of the educational material. Our work is committed to mitigating potential biases that could be introduced through the generative process and aims to promote the responsible development of LLMs for educational applications.

**Reprodicibility statement.** To ensure full reproducibility of our research, we will make all key components publicly available. This includes the complete EduDial dataset, the code used for data generation, the training scripts for EduDial-LLM 32B, and the source code for our 11-dimensional evaluation framework. We will also release the model weights for EduDial-LLM 32B, enabling other researchers to replicate our experiments and build upon our work. All resources will be hosted on a public repository upon publication.

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

# A APPENDIX

Table 5: Parameter Settings in SFT Stage for Four Models.

| Parameter | Teacher | Excellent Student | Medium Student | Struggling Student |
|---|---|---|---|---|
| Base Model | QwQ-32B-Preview | Qwen2.5-32B-Instruct | Qwen2.5-14B-Instruct | Qwen2.5-7B-Instruct |
| Learning Rate | 3e-4 | 5e-4 | 2e-4 | 5e-5 |
| Batch Size | 16 | 16 | 32 | 32 |
| Epochs | 2 | 3 | 5 | 5 |

## A.1 THE USE OF LARGE LANGUAGE MODELS (LLMS)

In this paper, we employ LLMs as an auxiliary tool. We utilized LLMs for language polishing to enhance clarity, and we manually reviewed all modifications.

## A.2 CASE STUDY

To illustrate the effectiveness of our approach, we present a comparative case study examining how EduDial-LLM and GPT-4o guide students in learning the concept of "Normal Distribution."

---

### Comparative Teaching Dialogue: Normal Distribution

**EduDial-LLM 32B**

**Teacher:** Students, today we're going to learn about a new statistical concept — normal distribution. Have you observed any patterns in the distribution of phenomena or data? For example, how do you think the test scores in your class might be distributed?

**Excellent Student:** I think there would probably be an average value, with most students' scores clustered around it, and only a few doing exceptionally well or poorly.

**Medium Student:** I think test scores would mostly be concentrated in the middle position, with possibly only a few students scoring particularly high or low.

**Struggling Student:** I'm not really sure, but I feel like some students would do well and some would do poorly.

**Teacher:** Very good! Your observations are very close to the normal distribution we're going to learn about today. Normal distribution is commonly used to describe many real-life phenomena, such as test scores, human height, weight, etc. Its characteristic is that most data is concentrated around the mean value, gradually decreasing on both sides. In our previous studies, we've already encountered the concepts of random variables and probability distributions. Do you remember what a probability distribution is?

**GPT-4o**

**Teacher:** Today, we are learning about normal distribution. Has anyone seen a "high in the middle, low on both sides" bell-shaped distribution in daily life?

**Excellent Student:** Yes! Test scores, long jump results, and height data all follow this pattern, with most people clustered around the average value.

**Medium Student:** Yes, class heights and test scores indeed follow this pattern, with fewer students getting very high or very low scores.

**Struggling Student:** I think I've heard that test scores are like this... But what does this have to do with normal distribution? Is it very complicated?

**Teacher:** Normal distribution can be represented by a bell-shaped curve, and its formula is:

$$\phi_{\mu,\sigma}(x) = \frac{1}{\sigma\sqrt{2\pi}} e^{-\frac{(x-\mu)^2}{2\sigma^2}}$$

where $\mu$ is the center position (mean), and $\sigma$ is the width of the distribution (standard deviation).

---

This comparison reveals significant pedagogical differences between the two approaches. EduDial-LLM employs an open-ended inquiry strategy that activates students' prior knowledge and encourages observation-based learning, progressively building from students' existing understanding to formal mathematical concepts. In contrast, GPT-4o follows a more traditional didactic approach, directly introducing the concept with pre-defined examples before immediately presenting the complex mathematical formula. Notably, EduDial-LLM demonstrates adaptive teaching by acknowledging all student responses and building upon each student's understanding level, while GPT-4o maintains a uniform teaching pace regardless of individual cognitive differences. Through this case study, we observe that EduDial-LLM's approach encourages deeper reflection and student engagement through exploratory questions, adapting its teaching strategy to provide appropriate scaffolding for different learner types, thereby embodying the differentiated instruction principles that are fundamental to effective mathematics education.

---

### Generation MTI Dataset Prompt Template

## STUDENT PROFILES

- **Excellent Student (s1):** {Excellent profile}
- **Medium Student (s2):** {Medium student profile}
- **Struggling Student (s3):** {Struggling student profile}

## TEACHER PROFILE

- **Teacher Profile:** {Teacher Profile}

## TEACHING STAGES & QUESTIONING STRATEGIES

1. **Introduction Stage**
   - *Scenario-based Questions:* Guide students to understand concepts through life experiences
     Example: {Scenario question example}
   - *Recall Questions:* Help students connect prior knowledge with new concepts
     Example: {Recall question example}

2. **Concept Exploration Stage**
   - *Prediction Questions:* Encourage students to predict new concepts based on existing knowledge
     Example: {Prediction question example}
   - *Verification Questions:* Verify conceptual understanding through examples
     Example: {Verification question example}

3. **Deep Understanding Stage**
   - *Zone of Proximal Development Questions:* Present questions slightly above students' current ability
     Example: {ZPD question example}
   - *Critical Thinking Questions:* Encourage students to question assumptions and explore new perspectives
     Example: {Critical thinking question example}

4. **Knowledge Application Stage**
   - *Integrative Questions:* Combine multiple knowledge points to solve complex problems
     Example: {Integrative question example}
   - *Open-ended Questions:* Present questions without unique answers to promote creative thinking
     Example: {Open-ended question example}

5. **Reflection and Summary Stage**
   - *Summary Questions:* Help students review and consolidate understanding
     Example: {Summary question example}
   - *Metacognitive Questions:* Guide students to evaluate their learning methods and effectiveness
     Example: {Metacognitive question example}

## TEACHING SYLLABUS

{Detailed teaching syllabus description}

## INTERACTION STRUCTURE

- At least two rounds of dialogue per phase
- Ensure coherence between phases

---

### Deep Exploration Strategy Prompt Template for Excellent Students

You are a mathematics teacher delivering a lesson based on the curriculum: {Teaching Content}
The classroom instruction is divided into five phases, with teacher (T) and student interactions in each phase:

- **T**: Represents the teacher, possessing systematic mathematical knowledge, rich teaching experience, and differentiated teaching abilities.
- **s1**: Represents high-performing students, typically with active thinking, solid foundation, and questions involving deep thinking or knowledge extension.
- **s2**: Represents average students, with relatively stable basic knowledge, but possibly insufficient mastery of problem-solving techniques and comprehensive application abilities.
- **s3**: Represents struggling students, who may have difficulty keeping up with classroom pace due to weak foundations or comprehension deficiencies.

## HIGH-PERFORMING STUDENT (S1) CHARACTERISTICS AND TEACHING STRATEGIES

### STUDENT PERFORMANCE CHARACTERISTICS:

1. **Question Types:**
   - **Inquiry questions**: e.g., "Why can derivatives represent the rate of change of a function? How is this conclusion derived?"
   - **Extension questions**: e.g., "Can this theorem be applied in more complex situations, such as non-linear equations?"
   - **Optimization questions**: e.g., "Is the standard answer to this problem unique? Is there a more concise solution?"

2. **Response Types:**
   - **Accurate answers with explanations**: e.g., "This function has an extreme point at x = 1 because the derivative is zero at that point and the second derivative is greater than zero, so it's a minimum value."
   - **Exploring conceptual essence**: e.g., "Besides the algebraic method, we can also use a geometric method to prove this conclusion."
   - **Creating knowledge connections**: e.g., "This problem is related to the arithmetic sequence we learned earlier, and can be solved using the general term formula for arithmetic sequences."

### TEACHER RESPONSE STRATEGY (DEEP EXPLORATION STRATEGY):

- Guide students to consider theoretical foundations, helping them understand the origins and deeper meanings of concepts
- Provide extended explanations or guide independent research, cultivating independent thinking abilities
- Encourage reflection on problems from different angles, exploring multiple solutions, fostering critical and innovative thinking
- Design higher-order challenging questions, encouraging application of learned knowledge to solve open-ended or comprehensive problems
- Guide students to reflect on problem-solving processes, promoting group discussion and collaboration

## TASK REQUIREMENTS

Design 25 non-repetitive single-round dialogues for high-performing students (s1) across five classroom phases, with 5 dialogues per phase. Each dialogue should include:

1. s1's statement (question or answer): Must conform to high-performing student characteristics, reflecting realistic teaching scenarios, limited to mathematical content
2. Teacher's response (T): Apply the deep exploration strategy, demonstrate teaching professionalism, and provide detailed guidance for student thinking processes

Generated dialogue format should be [s1], [T], ensuring rich, deep content that reflects the teacher's guidance for high-performing students' cognitive development.

---

---

### Thinking Guidance Strategy Prompt Template for Average Students

You are a mathematics teacher delivering a lesson based on the curriculum: {Teaching Content}
The classroom instruction is divided into five phases, with teacher (T) and student interactions in each phase:

- **T**: Represents the teacher, possessing systematic mathematical knowledge, rich teaching experience, and differentiated teaching abilities.
- **s1**: Represents high-performing students, typically with active thinking, solid foundation, and questions involving deep thinking or knowledge extension.
- **s2**: Represents average students, with relatively stable basic knowledge, but possibly insufficient mastery of problem-solving techniques and comprehensive application abilities.
- **s3**: Represents struggling students, who may have difficulty keeping up with classroom pace due to weak foundations or comprehension deficiencies.

## AVERAGE STUDENT (S2) CHARACTERISTICS AND TEACHING STRATEGIES

### STUDENT PERFORMANCE CHARACTERISTICS:

1. **Question Types:**
   - **Application questions**: e.g., "In this problem, how do I determine whether to use the binomial theorem or direct expansion?"
   - **Skill-based questions**: e.g., "Why must this problem be simplified first rather than directly substituting into the formula?"
   - **Detail-oriented questions**: e.g., "Why can this transformation step be done this way?"

2. **Response Types:**
   - **Partially correct answers**: e.g., "For the equation $x^2-4=0$, after moving terms and taking the square root, we get x=2."
   - **Expressing uncertainty**: e.g., "I think I should use the formula to solve it, but I don't know how to proceed to the next step."
   - **Seeking hints**: e.g., "This problem is a bit difficult, could you give me another hint?"

### TEACHER RESPONSE STRATEGY (THINKING GUIDANCE STRATEGY):

- Help students understand internal connections between knowledge points, clarifying their functions and applicable scenarios
- Gradually explain the logic of problem-solving processes, encourage independent restatement of problem-solving ideas, cultivating thinking abilities
- Provide in-depth explanations for detail-oriented questions, offer efficient practice strategies to help consolidate knowledge application
- Provide positive feedback, promptly praise progress and correct portions, offering improvement suggestions in an encouraging manner
- Provide layered guidance, breaking down problem-solving processes into smaller steps, guiding completion step-by-step

## TASK REQUIREMENTS

Design 25 non-repetitive single-round dialogues for average students (s2) across five classroom phases, with 5 dialogues per phase. Each dialogue should include:

1. s2's statement (question or answer): Must conform to average student characteristics, reflecting realistic teaching scenarios, limited to mathematical content
2. Teacher's response (T): Apply the thinking guidance strategy, demonstrate teaching professionalism, and provide detailed guidance for problem-solving approaches

Generated dialogue format should be [s2], [T], ensuring responses are inspirational and effectively help average students understand knowledge points and improve problem-solving abilities.

---

**Confidence Building Strategy Prompt Template for Struggling Students**

You are a mathematics teacher delivering a lesson based on the curriculum: {Teaching Content}
The classroom instruction is divided into five phases, with teacher (T) and student interactions in each phase:

- **T**: Represents the teacher, possessing systematic mathematical knowledge, rich teaching experience, and differentiated teaching abilities.
- **s1**: Represents high-performing students, typically with active thinking, solid foundation, and questions involving deep thinking or knowledge extension.
- **s2**: Represents average students, with relatively stable basic knowledge, but possibly insufficient mastery of problem-solving techniques and comprehensive application abilities.
- **s3**: Represents struggling students, who may have difficulty keeping up with classroom pace due to weak foundations or comprehension deficiencies.

### STRUGGLING STUDENT (S3) CHARACTERISTICS AND TEACHING STRATEGIES

#### STUDENT PERFORMANCE CHARACTERISTICS:

1. **Question Types:**
   - **Fundamental questions**: e.g., "Why are alternate interior angles equal when two parallel lines are crossed by a transversal?"
   - **Understanding questions**: e.g., "I'm completely lost, where should I start with this problem?"
2. **Response Types:**
   - **Incorrect answers**: e.g., "Teacher, the expansion of (a+b)² is 2a+2b."
   - **Vague responses**: e.g., "I think I calculated something wrong, I'm not sure if the answer is correct."
   - **Direct expression of inability**: e.g., "Teacher, I don't know how to do this."

#### TEACHER RESPONSE STRATEGY (CONFIDENCE BUILDING STRATEGY):

- Emphasize repeated explanation of basic knowledge, using straightforward language to aid understanding
- Patiently guide step-by-step analysis of problems, avoiding presenting too much information at once
- Use simple, concrete examples to help build confidence and reduce frustration
- Affirm students' efforts and correct portions, then help gradually correct thinking paths, protecting self-confidence
- Start from the most basic elements, allowing students to experience step-by-step success, reducing psychological burden

### TASK REQUIREMENTS

Design 25 non-repetitive single-round dialogues for struggling students (s3) across five classroom phases, with 5 dialogues per phase. Each dialogue should include:

1. s3's statement (question or answer): Must conform to struggling student characteristics, reflecting realistic teaching scenarios, limited to mathematical content
2. Teacher's response (T): Apply the confidence-building strategy, demonstrating teaching patience and understanding of student psychology

Generated dialogue format should be [s3], [T], ensuring responses are clear, progressive, and help struggling students build confidence in mathematics learning.

---

**Evaluation Prompt Template**

Please evaluate the following teaching content based on 11 teaching indicators spanning two dimensions: overall quality and content quality. Each overall quality indicator and relevance score ranges from 1-5 points (1 being the lowest, 5 being the highest). Coverage is rated from 0%-100% (0% being the lowest, 100% being the highest).

## ASSESSMENT OBJECT

Teaching content: {Teaching content}
Teaching outline: {Teaching outline}

## OVERALL QUALITY DIMENSION (1-5 POINTS EACH)

### 1. INSIGHT

**Definition:** Assessing whether the teacher can accurately capture and deeply understand students' learning needs, knowledge levels, and question intentions.
**Assessment points:**

- **Need identification:** Whether the teacher identifies students' specific learning needs and difficulties through questioning, observation, or other methods
- **Problem analysis:** Whether the teacher can analyze students' questions and understand the learning obstacles or misconceptions behind them
- **Personalized understanding:** Whether the teacher demonstrates understanding of each student's unique learning style and needs

**Score:** [ ]/5

### 2. RESPONSE

**Definition:** Assessing whether the teacher can effectively solve students' problems and provide practical and constructive guidance.
**Assessment points:**

- **Problem solving:** Whether the teacher can provide clear and effective solutions to students' problems
- **Specificity of guidance:** Whether the advice provided is specific, actionable, and can help students improve in practice
- **Resource provision:** Whether the teacher recommends relevant resources (such as textbooks, exercises, reference materials) to support students' further learning

**Score:** [ ]/5

### 3. FEEDBACK

**Definition:** Assessing whether the teacher can provide timely and effective feedback to help students improve their learning.
**Assessment points:**

- **Timeliness:** Whether feedback is provided promptly after students raise questions, avoiding delays in students' learning progress
- **Constructiveness:** Whether the feedback content is constructive and can guide students on how to improve
- **Two-way communication:** Whether the teacher encourages students to respond to feedback, promoting two-way communication

**Score:** [ ]/5

### 4. THINKING

**Definition:** Assessing whether the teacher can stimulate students' critical thinking abilities, including analytical ability, open-mindedness, and self-assessment ability.
**Assessment points:**

- **Question guidance:** Whether the teacher guides students to think deeply through open-ended questions
- **Analysis training:** Whether the teacher cultivates students' analytical abilities, helping them break down complex problems

- **Self-assessment:** Whether the teacher encourages students to reflect and self-assess to enhance independent learning abilities

**Score:** [ ]/5

## 5. INTERACTIVITY

**Definition:** Assessing the frequency and quality of teacher-student interactions in teaching dialogues, including questioning techniques and ways of responding to student feedback.
**Assessment points:**

- **Questioning techniques:** Whether the teacher uses effective questioning methods to promote student thinking and participation
- **Response to feedback:** Whether the teacher actively responds to student feedback, promoting continued interaction
- **Interaction frequency:** Whether interactions between teacher and students are frequent, avoiding one-way knowledge transmission

**Score:** [ ]/5

## 6. EMOTIONAL SUPPORT

**Definition:** Assessing whether the teacher can provide emotional support during the teaching process, establish a positive learning environment, and help students build confidence and positive learning attitudes.
**Assessment points:**

- **Emotional care:** Whether the teacher pays attention to students' emotional states and provides necessary care and support
- **Motivation and encouragement:** Whether the teacher stimulates students' learning motivation and self-confidence through encouragement and praise
- **Building trust:** Whether the teacher establishes trust relationships with students, making them feel safe and respected

**Score:** [ ]/5

## 7. ADAPTABILITY

**Definition:** Assessing the teacher's ability to adjust teaching methods according to different students' learning styles and needs, ensuring that each student receives personalized guidance.
**Assessment points:**

- **Teaching method adjustment:** Whether the teacher adjusts teaching strategies and methods based on student feedback and learning progress
- **Personalized guidance:** Whether the teacher can provide personalized guidance and support based on different students' characteristics
- **Flexible response:** Whether the teacher can flexibly respond to unexpected situations in the classroom to ensure teaching effectiveness

**Score:** [ ]/5

## 8. FLUENCY

**Definition:** Assessing whether the teacher's expression is clear, easy to understand, and natural in tone.
**Assessment points:**

- **Language expression:** Whether the language used by the teacher is accurate and concise, avoiding ambiguous or obscure expressions
- **Logical structure:** Whether the teacher's explanation has a clear logical structure that facilitates student understanding
- **Natural tone:** Whether the teacher's tone in communication is friendly and patient, creating a good communication atmosphere

**Score:** [ ]/5

### 9. GOAL

**Definition:** Assessing whether the teacher's guidance helps achieve predetermined teaching objectives, such as knowledge mastery, skill development, etc.
**Assessment points:**

- **Clear objectives:** Whether teaching activities have clearly set specific learning objectives

- **Goal alignment:** Whether the teacher's teaching behaviors and guidance align with the set objectives

- **Outcome assessment:** Whether the teacher uses assessment methods to verify if students have achieved the predetermined objectives

**Score:** [ ]/5

## CONTENT QUALITY DIMENSION

### 10. RELEVANCE

**Definition:** Assessing whether the teaching content is relevant to the teaching outline of this lesson.
**Assessment points:**

- Whether the content aligns with the themes and objectives of the teaching outline

- Whether the content includes core concepts and knowledge points related to the course topic

**Score:** [ ]/5

### 11. COVERAGE

**Definition:** Assessing the proportion of teaching outline knowledge points covered by the teaching content.
**Assessment points:**

- The proportion of knowledge points in the teaching outline covered by the teaching content

- Whether the core concepts and key knowledge points in the teaching outline are covered

**Coverage rate:** [ ]%

## SUMMARY EVALUATION

[Provide comprehensive evaluation and improvement suggestions here]

Table 6: Pedagogical Questioning Strategies for the Five-Stage Teaching Process.

| Question Type | Purpose | Cognitive Process | Exemplar Questions |
|---|---|---|---|
| Scenario-based | Connect abstract concepts to real-world applications | Activation of prior knowledge $\rightarrow$ Contextualization | "In planning a 'Community Garden Project' with varied participant skills (vegetable growing, design, tool provision), how would you optimize team allocation?" |
| Retrospective | Bridge previous and new knowledge | Memory retrieval $\rightarrow$ Knowledge integration | "How does today's set theory connect with our earlier classification of numbers?" |
| Predictive | Stimulate hypothesis formation | Pattern recognition $\rightarrow$ Logical conjecture | "Predict the elements when grouping all even numbers from 1 to 10 into a set" |
| Verification | Develop critical evaluation skills | Hypothesis testing $\rightarrow$ Counterexample analysis | "If we add 0 to the positive integer set, does it remain a positive integer set? Justify." |
| Zone of Proximal Development | Scaffold conceptual advancement | Guided discovery $\rightarrow$ Generalization | "Extend the 1-10 even number set to formulate a general representation for any range" |
| Critical | Foster alternative perspectives | Divergent thinking $\rightarrow$ Justification | "Design a novel integer classification beyond odd-/even with explanatory principles" |
| Integrative | Synthesize multiple concepts | Cross-domain application $\rightarrow$ System thinking | "Find a number in $\mathbb{Q}$ but not $\mathbb{Z}$ and explain its significance" |
| Open-ended | Encourage creative exploration | Speculative reasoning $\rightarrow$ Perspective-taking | "How would a mathematical system with only $\mathbb{Z}$ and irrationals impact real-world applications?" |
| Summative | Consolidate learning outcomes | Knowledge organization $\rightarrow$ Key point identification | "Summarize the core concepts of set representation covered today" |
| Metacognitive | Develop self-regulated learning | Process reflection $\rightarrow$ Strategy planning | "Which aspects do you feel confident about? What improvement plan would address gaps?" |

Table 7: Role Profiles for Simulating Authentic Teaching Scenarios.

| Dimensions | Teacher | Excellent Student | Medium Student | Struggling Student |
|---|---|---|---|---|
| Cognitive Ability | Rigorous understanding of mathematical concepts; capable of clear and accurate explanations | Sharp logical thinking; quickly comprehends new content; applies knowledge to solve complex problems | Good comprehension but requires more time to absorb new knowledge; capable of understanding core content | Weak foundational knowledge; difficulties grasping new concepts; struggles with complex problem-solving |
| Learning Motivation | Committed to comprehensive knowledge transfer; provides emotional support to enhance student confidence | Strong curiosity for knowledge; explores mathematics beyond textbooks; seeks interdisciplinary connections | Moderate interest in learning; requires occasional encouragement to maintain engagement | Lacks interest in mathematics; easily frustrated when unable to keep pace; diminished learning drive |
| Classroom Participation | Facilitates active discussions; creates an inclusive learning environment; provides real-time feedback | Active participant in discussions; assumes leadership roles in group learning; helps peers understand difficult concepts | Moderate participation level; contributes when prompted; neither overly active nor passive | Easily distracted; minimal voluntary participation; requires significant prompting to engage |
| Learning Strategies | Employs diverse teaching methodologies; adjusts explanations to accommodate different learning levels; uses varied assessment techniques | Self-regulated learning; effectively balances academic and extracurricular activities; employs multiple problem-solving approaches | Limited range of learning strategies; relies on note-taking and peer discussion; benefits from structured learning frameworks | Highly dependent on external support; lacks independent problem-solving skills; requires step-by-step guidance |
| Self-Regulation | Monitors classroom dynamics; adapts teaching pace and content based on student responses; balances coverage with comprehension | Self-disciplined; maintains consistent learning motivation despite challenges; sets clear goals with actionable implementation plans | Requires external structure; occasionally lacks confidence, especially with complex problems; benefits from incremental successes | Strong dependency on external motivation; easily discouraged by failures; shows improvement potential with personalized guidance and encouragement |

Table 8: Teaching Strategies for Different Student Question and Reply Types.

| Student Behavior | Teacher Strategy |
|---|---|
| **Excellent Students** | **Depth Exploration Strategy** |
| Inquiry Questions | Guide students to understand the concept essence |
| Extension Questions | Guide independent research and thinking |
| Optimization Questions | Compare solutions from multiple perspectives |
| Accurate answers with explanations | Design advanced questions for deeper understanding |
| Exploring essence | Explore core elements and mechanisms |
| Connecting relevant knowledge | Establish connections across contexts |
| **Medium Students** | **Guided Thinking Strategy** |
| Application Questions | Help understand knowledge connections |
| Technique Questions | Explain logic and techniques of problem-solving |
| Detail Questions | Analyze inquiries to consolidate knowledge |
| Partially correct answers | Affirm correct portions, provide feedback |
| Expressing uncertainty | Break complex problems into sub-steps |
| Seeking hints | Guide analysis of key components |
| **Struggling Students** | **Confidence Building Strategy** |
| Basic Questions | Emphasize fundamentals with simple methods |
| Comprehension Questions | Guide problem analysis for understanding |
| Learning Efficacy Questions | Use examples to build confidence |
| Incorrect answers | Provide encouragement and gradual guidance |
| Vague answers | Use prompts to clarify reasoning |
| Unable to answer | begin with basics to reduce pressure |