# OpenReview forum: "EduDial: Constructing a Large-scale Multi-turn Teacher–Student Dialogue Corpus"
_ICLR.cc/2026/Conference — ICLR 2026 Conference Withdrawn Submission_

### Official Review · Reviewer_AVdn · 2025-10-27

**Soundness:** 2
**Presentation:** 2
**Contribution:** 3
**Rating:** 4
**Confidence:** 2

**Summary:**

This article presents a high-quality multi-round teacher-student dialogue dataset (EduDial) that covers 345 core K-12 mathematics knowledge points and contains 34,250 dialogues. The dataset is split into two parts:
(1) MTI (SFT) dataset – 13,700 “one-teacher-three-students” dialogues generated by GPT-o1 under five teaching principles;
(2) PDTS (DPO) dataset – 20,500 “one-teacher-one-student” preference pairs generated by an SFT model and further refined with differentiated-teaching prompts.
All samples are validated through Expert-Machine Dual Verification. An 11-dimensional expert-designed evaluation framework is proposed and used to benchmark mainstream LLMs. After two-stage SFT+DPO training on EduDial, the 32B model surpasses current state-of-the-art LLMs.

**Strengths:**

(1) Practical educational value: the dataset is constructed under expert pedagogical guidance and emphasizes personalized instruction, emotional support, and teaching students according to their aptitude.
(2) Comprehensive evaluation: a multidimensional framework is established and applied to a wide range of open-source and proprietary LLMs, providing a solid foundation for follow-up research.
(3) Rigorous quality control: expert verification and machine filtering are adopted throughout data generation.

**Weaknesses:**

(1) Evaluation protocol ambiguity: it is unclear whether the quality scores come from the 1v3 (MTI) setting, the 1v1 (PDTS) setting, or both; the score differences between Table 2 and Table 3 are not explained.
(2) Student-model issue: the paper mentions that the student side of the SFT dataset receives inputs from the teacher and other students, yet it does not clarify why an SFT-trained “student model” is used or how it participates in evaluation; if the evaluated teacher model interacts with the same student model it was trained with, the reported gains may reflect familiarity rather than generalizable teaching ability.
(3) OOD generalization concern: because student profiles are identically distributed in training and test sets, performance improvements might degrade for out-of-distribution students.
(4) Stage-split motivation: the paper does not adequately justify why dataset construction and training are split into two distinct stages (1v3 vs. 1v1); the relevance of the unused stage to the final evaluation scenario needs further explanation.
(5) Differentiated teaching location: differentiated instruction is emphasized in the DPO dataset but is absent from the SFT dataset; the rationale for this asymmetry is not provided.
(6) Dialogue length: the maximum sequence length in the DPO stage is set to 1024 tokens, which may be insufficient for longer multi-turn conversations.

**Questions:**

See Weaknesses.

---

> ### Author Response · Authors · 2025-11-21
>
> > **Q1:** Evaluation protocol setting.
>
> **A1:** We thank the reviewer for this comment. We clarify that both the MTI and PDTS datasets are designed to serve one-to-many classroom scenarios. While PDTS is constructed as a one-to-one preference dataset, its purpose is to enable the teacher model to dynamically adjust teaching strategies for students with different cognitive levels in one-to-many scenarios.
>
> Table 2 evaluates one-to-many teaching, where the teacher model simultaneously interacts with three students of different cognitive levels, with dialogues ending upon completing the teaching or reaching 15 rounds. Table 2 results demonstrate our model's excellent performance in one-to-many scenarios. To verify the model's effectiveness across different scenarios, Table 3 presents an ablation study in one-to-one teaching, where the teacher model interacts with a single student, also limited to 15 rounds. Both tables employ the same evaluation dimensions and methods. Table 3 results confirm our model also performs excellently in one-to-one scenarios.
>
> > **Q2:** Student-model issue.
>
> **A2:** Thank you for your comment. We wish to clarify the following:
>
> Regarding the student model, our experiments show that when we prompt the model to play the role of a student, it drifts from the intended behavior after 4-5 rounds of conversation. However, with SFT fine-tuning, the student model maintains stable behavior across multiple interactions.
>
> Regarding student model participants in evaluation, the student model engages in multi-round dialogues with the teacher model, and the resulting dialogues form the test set, which is used to evaluate the teacher model's teaching effectiveness.
>
> Regarding the generalization issue, we respectfully disagree with your point: first, the student model learns "how to behave like a student." Second, during the evaluation phase, the teacher model teaches new knowledge that was not encountered during training, which demonstrates its true ability to generalize. Furthermore, the model's excellent performance in one-on-one teaching scenarios (see Table 3) further proves its ability to generalize in different teacher environments.
>
>
> > **Q3:** OOD generalization concern.
>
> **A3:** To validate our model's generalization capability on out-of-distribution students, we conducted an additional ablation study. We employed GPT-4o as the student model and constructed six student types based on the five cognitive dimensions defined in SocraticLM (Student Types 1-5 demonstrate poor performance in problem understanding, instruction understanding, calculation ability, knowledge mastery, and thirst for learning, respectively; Student Type 6 excels in all dimensions). Using the original prompts from the SocraticLM paper, we had GPT-4o role-play these students in interactions with EduDial-LLM. The results demonstrate that our model maintains high teaching quality when interacting with out-of-distribution students, achieving an average score of 4.36, only 0.12 points lower than the in-distribution students in Table 3, thereby confirming the model's strong generalization capability.
>
> | Model | INS | RES | FB | THK | INT | EMO | ADP | FLU | GOL | AVG |
> |-------|-----|-----|-----|-----|-----|-----|-----|-----|-----|-----|
> | Student Type 1: Poor Problem Understanding | 4.15 | 4.38 | 4.52 | 4.19 | 4.22 | 4.08 | 4.65 | 4.61 | 4.73 | 4.28 |
> | Student Type 2: Poor Instruction Understanding | 4.21 | 4.42 | 4.48 | 4.16 | 4.27 | 4.11 | 4.58 | **4.92** | 4.79 | 4.38 |
> | Student Type 3: Poor Calculation | 4.19 | 4.45 | 4.56 | 4.22 | 4.31 | 4.15 | 4.71 | 4.68 | 4.81 | 4.36 |
> | Student Type 4: Poor Knowledge Mastery | 4.12 | **4.47** | 4.49 | 4.14 | 4.19 | 4.06 | **4.73** | 4.65 | 4.76 | 4.29 |
> | Student Type 5: Poor Thirst for Learning | 4.18 | 4.41 | 4.54 | 4.11 | 4.26 | **4.17** | 4.62 | 4.72 | 4.78 | 4.31 |
> | Student Type 6: Excellent at All Dimensions | **4.25** | 4.46 | **4.63** | **4.28** | **4.38** | 4.14 | 4.69 | 4.74 | **4.87** | **4.42** |
>
>
> > **Q4:** Stage-split motivation.
>
> **A4:** We wish to clarify that the stage-split design stems from the fact that these two teaching capabilities require different learning approaches. The five-stage teaching process is a fixed instructional procedure that can be mastered through SFT training. In contrast, differentiated teaching strategies require learning through DPO's contrastive data (preferred vs. rejected responses) to discern how to make better choices. If both are trained simultaneously using only SFT, the model can only observe demonstration data and cannot effectively acquire strategy selection capabilities.
>
> > **Q5:** Differentiated teaching location.
>
> **A5:** This concern has been addressed in our response to Q4.
>
> > **Q6:** Dialogue length.
>
> **A6:** We clarify that the DPO stage processes single-turn contrastive samples (i.e., individual <instruction, chosen response, rejected response> triplets), which average under 1000 tokens per sample.

---

> ### Comment · Reviewer_AVdn · 2025-11-26
>
> Thank you for your response. The author's experimental design for OOD addresses my concerns. The explanation of the stage-split approach is generally acceptable (though it should be clearly articulated in the paper, including the evaluation protocol). I concur with Reviewer 1cck's comment that the interaction style appears somewhat unnatural. However, the introduction of this scenario itself contributes. After considering all factors, I decide to increase the score.

---

### Official Review · Reviewer_1cck · 2025-11-01

**Soundness:** 3
**Presentation:** 3
**Contribution:** 3
**Rating:** 4
**Confidence:** 3

**Summary:**

This paper presents EduDial, a comprehensive multi-turn teacher-student dialogue dataset. EduDial covers 345 core knowledge points and consists of 34,250 dialogue sessions generated through interactions between teacher and student agents. The dataset is designed to help large language models (LLMs) ask appropriate questions at the right time in educational settings. It follows five progressive teaching stages, introduction, concept exploration, deep understanding, knowledge application, and reflection, and incorporates ten questioning strategies aimed at students with varying cognitive levels. The authors also train EduDial-LLM 32B using this dataset and evaluate the model using an 11-dimensional evaluation framework, assessed by both humans and LLM-based evaluators.

**Strengths:**

1. Propose a multi-turn teacher-student dialogue dataset that covers 345 core knowledge points and comprises 34,250 dialogue sessions.
2. Design teacher and student agents that follow progressive teaching stages and reflect students’ varying cognitive levels.
3. Conduct manual reviews to assess both the quality of the dataset and the teaching effectiveness of the model.

**Weaknesses:**

1. It is unclear whether the MTI and DPO datasets are intended to simulate classroom interactions involving multiple students and a single teacher, or one-on-one teacher-student dialogues. For example, line 344 suggests a one-on-one setting, while the dialogue examples in the appendix appear to reflect one-to-many interactions. In the latter case, it is also unclear whether the dialogues are specifically tailored to each student's cognitive level.
2. It is not evident whether the quality of the student agent simulation has been evaluated.
3. While the paper mentions a human evaluation, it lacks information about the qualifications or expertise of the evaluators, specifically, whether they are capable of assessing teacher-student educational dialogues.

**Questions:**

Is there an ablation study or justification for choosing QwQ-32B-Preview as the backbone model?

---

> ### Author Response · Authors · 2025-11-21
>
> > **Q1:** Are the MTI and DPO datasets intended to simulate multi-student classroom scenarios or one-on-one teaching, and are the dialogues tailored to each student's cognitive level?
>
> **A1:** Thank you for your suggestion. We aim to simulate multi-student classroom scenarios. MTI contains multi-turn dialogues between a teacher and multiple students. PDTS extends MTI as a preference dataset for differentiated instruction. We characterize three student profiles (excellent, medium, struggling) through five dimensions: Cognitive Ability, Learning Motivation, Classroom Participation, Learning Strategies, and Self-Regulation (Table 7, line 1165). Although PDTS construction involves generating differentiated responses for individual students (line 344), the purpose is to enable the model to identify different students (excellent, medium, struggling) in classroom settings and provide corresponding teaching strategies. Through training on the data, the model can dynamically adjust teaching methods based on each student's performance in classroom interactions.
>
> > **Q2:** The quality of the student agent simulation.
>
> **A2:** To evaluate the quality of student simulations, we supplement with an ablation study. Five experts independently rated the responses of three student types in the test set across the five dimensions we defined (Table 7), assessing the alignment between simulated behaviors and predefined profiles. Ratings based on a 5-point Likert scale (1=misaligned, 5=fully aligned). As shown in the table below, the three student types achieved scores ranging from 3.82 to 4.76 across the five dimensions, with an overall average score of 4.48, demonstrating high consistency between simulated behaviors and their respective profiles. Additionally, we measured inter-rater reliability using Cohen's Kappa coefficient. We calculated the Kappa coefficient for each pairwise combination of experts and averaged the results, yielding a value of 0.71, indicating substantial agreement among expert evaluations.
>
> | Student Profile | Cognitive Ability | Learning Motivation | Classroom Participation | Learning Strategies | Self-Regulation | Average |
> |-----------------|-------------------|---------------------|-------------------------|---------------------|-----------------|---------|
> | Excellent Student | **4.58** | **4.76** | **4.68** | **4.52** | 4.73 | **4.65** |
> | Medium Student | 4.43 | 4.62 | 3.96 | 4.48 | 4.58 | 4.41 |
> | Struggling Student | 4.47 | 4.71 | 3.82 | 4.45 | **4.76** | 4.44 |
> | Average | 4.49 | 4.69 | 4.15 | 4.48 | 4.69 | 4.48 |
>
> > **Q3:** The human evaluators' qualifications.
>
> **A3:** Thank you for your comment. We wish to clarify that all evaluators in our study are experienced mathematics teachers. They possess a professional understanding of teacher-student dialogues and student learning behaviors, making them qualified to assess the quality of the data.
>
> > **Q4:** Selection of the backbone model.
>
> **A4:** We thank the reviewer for this valuable suggestion. To address this concern, we supplement with an ablation study.
>
> First, we selected QwQ-32B-Preview as our primary backbone model because it achieves the best overall performance (AVG score of 4.47) after training with our data. To validate this choice, we conducted comprehensive ablation experiments on Llama-3.3-70B (i.e., Llama-3.3-70B-Instruct), DeepSeek-32B (i.e., DeepSeek-R1-Distill-Qwen-32B), Qwen2.5-72B (i.e., Qwen2.5-72B-Instruct), and Qwen2.5-Math (i.e., Qwen2.5-Math-72B) in a unified environment using the same configuration as described in the paper. The results show that QwQ-32B-Preview, as the backbone model, outperforms these competitive models.
>
> Second, the ablation experiments also demonstrate the generalizability of our data. Models with different architectures and scales all achieve significant performance improvements after training.
>
> | Model | INS | RES | FB | ThK | INT | EMO | ADP | FLU | GOL | AVG |
> |-------|-----|-----|-----|-----|-----|-----|-----|-----|-----|-----|
> | Llama-3.3-70B | 2.56 | 3.08 | 2.53 | 2.36 | 2.81 | 2.72 | 2.47 | 4.25 | 3.56 | 2.93 |
> | Llama-3.3-70B w/ SFT w/ DPO | 3.83 | 4.29 | 3.96 | 4.10 | 4.02 | 4.12 | 4.24 | **4.64** | 4.58 | 4.15 |
> | DeepSeek-32B | 2.06 | 2.08 | 1.26 | 2.08 | 1.58 | 1.44 | 1.49 | 3.50 | 3.02 | 2.06 |
> | DeepSeek-32B w/ SFT w/ DPO | 3.93 | 3.92 | 3.81 | 4.00 | 3.98 | 4.01 | 4.02 | 4.47 | 4.50 | 4.07 |
> | Qwen2.5-72B | 2.89 | 2.74 | 2.43 | 2.89 | 2.78 | 2.67 | 2.41 | 3.62 | 3.39 | 2.87 |
> | Qwen2.5-72B w/ SFT w/ DPO | 4.32 | 4.49 | 4.37 | 4.41 | 4.54 | 4.13 | 4.61 | 4.48 | 4.59 | 4.44 |
> | Qwen2.5-Math | 2.02 | 2.32 | 1.96 | 1.93 | 1.91 | 1.89 | 1.84 | 2.84 | 2.70 | 2.16 |
> | Qwen2.5-Math w/ SFT w/ DPO | 4.58 | 4.08 | 4.49 | 4.22 | 4.58 | 4.08 | 4.38 | 4.24 | 4.27 | 4.32 |
> | QwQ-32B-Preview | 2.89 | 3.24 | 2.65 | 2.43 | 2.69 | 2.67 | 2.16 | 3.15 | 2.89 | 2.75 |
> | **EduDial-LLM (Ours)** | **4.58** | **4.12** | **4.62** | **4.55** | **4.64** | **4.10** | **4.60** | 4.43 | **4.56** | **4.47** |

---

> > ### Comment · Reviewer_1cck · 2025-11-25
> >
> > Thank you for the clarification. Most of the concerns have been resolved. However, there is still some uncertainty about how the teacher can effectively respond to students with different understanding levels in a multi-turn dialogue between a teacher and multiple students setting. Based on the provided dialogue examples, the teacher speaks once, each student responds separately, and then the teacher provides one reply addressing all students’ responses together, repeating this pattern. This interaction style seems somewhat unnatural compared to real classroom dynamics, where teachers typically respond interactively and adaptively to students one at a time rather than addressing multiple students with varying comprehension levels simultaneously. As a result, it appears difficult to objectively assess whether a single teacher response can appropriately satisfy students representing three distinct understanding profiles. This could easily introduce bias and reduce evaluation validity. For these reasons, I will maintain my original assessment.

---

### Official Review · Reviewer_A2ik · 2025-11-02

**Soundness:** 2
**Presentation:** 2
**Contribution:** 2
**Rating:** 2
**Confidence:** 3

**Summary:**

This paper aims to address the issue that general dialogue benchmarks fail to evaluate the core capabilities of large language models (LLMs) in the education domain. It proposes EduDial, a multi-turn teacher-student dialogue dataset, which covers 345 core knowledge points and 34,250 dialogue sessions. Guided by Bloom’s taxonomy, the dataset is designed with a five-stage teaching process and ten questioning strategies, and differentiated teaching strategies are formulated for students at different cognitive levels. Based on EduDial, the researchers trained EduDial-LLM 32B and proposed an 11-dimensional evaluation framework.

**Strengths:**

The paper presents a good approach. The proposed dialogue dataset may be beneficial for certain scenarios in the education domain.
The figures in this paper clearly illustrate the entire process, allowing for quick comprehension of the workflow.

**Weaknesses:**

1. The paper mentions two major challenges: (1) Frequent questioning may disrupt students’ cognitive flow; (2) Adjusting questioning strategies according to different teaching stages. However, the paper does not adequately elaborate around these two challenges. I fail to understand how the work in this paper addresses the first challenge.

2. In the validation section, the use of the SFT + DPO verification method to validate the effectiveness of the dataset is questionable. How can I tell whether the improved performance stems from well-trained models or the superiority of the dataset? This remains doubtful.

3. I do not know what the base model used for training EduDial is—did I miss any relevant information?

4. The discussion on the combined evaluation of human expert assessment and machine evaluation is far from sufficient. Although an experiment proves a certain degree of similarity between human and machine evaluations, it cannot rule out the bias in human expert assessment.

5. Relying solely on the expert-machine dual verification (GPT-4o + expert review) and lacking data calibration involving real teachers and students, the paper fails to prove that the data can truly reflect classroom teaching logic.

**Questions:**

Please see weakness above

---

> ### Author Response · Authors · 2025-11-21
>
> > **Q1:** Challenges are inadequately elaborated. How does the paper address Challenge 1?
>
> **A1:** We respectfully disagree with the reviewer's concern that the challenges are insufficiently articulated. We clearly articulated two core challenges in the third paragraph of the introduction.
>
> Regarding "how to address Challenge 1": it concerns determining optimal timing for questioning. The paper exemplifies this through the statement "when students are still grasping a new concept, frequent questioning may interfere with their cognitive flow," which reflects how premature and frequent questioning before knowledge mastery exacerbates confusion and represents inappropriate timing. Our solution (Sections 3.2 and 3.3): We construct a five-stage progressive teaching process based on Bloom's Taxonomy. Each stage represents an appropriate timing for questioning, because once students acquire the cognitive abilities of that stage, questioning effectively guides them to higher cognitive stages. Meanwhile, we designed differentiated teaching strategies to provide timely support when students express confusion, ensuring they have mastered the current stage before advancement.
>
> > **Q2:**  Validate the effectiveness of the dataset.
>
> **A2:** To further clarify the effectiveness of our training corpora, we supplement with an ablation study. Results show: (1) SFT Data Quality: Compared to the base model QwQ-32B-Preview (AVG 2.75), QwQ-32B-Preview w/ SocraticLM[1], and QwQ-32B-Preview w/ MTI achieve 3.69 and 4.23, respectively, confirming MTI's superiority over other teaching SFT datasets. (2) DPO Data Quality: QwQ-32B-Preview w/ MTI w/ PDTS-Reversed scores 4.18, lower than QwQ-32B-Preview w/ MTI (4.23), while QwQ-32B-Preview w/ MTI w/ PDTS improves to 4.47, validating PDTS effectiveness. (3) Cross-dataset Robustness: QwQ-32B-Preview w/ SocraticLM w/ PDTS improves by 8.4% over their respective SFT baselines, while the PDTS-Reversed versions only gain 1.9%, further confirming our preference for dataset quality.
>
> | Model | INS | RES | FB | THK | INT | EMO | ADP | FLU | GOL | AVG |
> |-------|-----|-----|-----|-----|-----|-----|-----|-----|-----|-----|
> | **Base Model** |
> | QwQ-32B-Preview | 2.89 | 3.24 | 2.65 | 2.43 | 2.69 | 2.67 | 2.16 | 3.15 | 2.89 | 2.75 |
> | **Base Model + SFT** |
> | QwQ-32B-Preview w/ SocraticLM | 3.64 | 3.71 | 3.68 | 3.56 | 3.63 | 3.58 | 3.65 | 3.98 | 3.76 | 3.69 |
> | QwQ-32B-Preview w/ MTI | 4.12 | 4.35 | 4.28 | 3.88 | 4.14 | 3.90 | 4.13 | 4.79 | 4.49 | 4.23 |
> | **Base Model + SFT + DPO (PDTS-Reversed)** |
> | QwQ-32B-Preview w/ SocraticLM w/ PDTS-Reversed | 3.71 | 3.78 | 3.75 | 3.62 | 3.70 | 3.65 | 3.72 | 4.05 | 3.83 | 3.76 |
> | QwQ-32B-Preview w/ MTI w/ PDTS-Reversed | 4.07 | 4.30 | 4.23 | 3.83 | 4.09 | 3.85 | 4.08 | 4.74 | 4.44 | 4.18 |
> | **Base Model + SFT + DPO (PDTS)** |
> | QwQ-32B-Preview w/ SocraticLM w/ PDTS | 3.95 | 4.02 | 4.00 | 3.87 | 3.94 | 3.89 | 3.96 | 4.29 | 4.07 | 4.00 |
> | **EduDial-LLM (Ours)** | **4.58** | **4.12** | **4.62** | **4.55** | **4.64** | **4.10** | **4.60** | **4.43** | **4.56** | **4.47** |
>
> All experiments were conducted using the same configuration as in the paper. PDTS-Reversed swaps the chosen and rejected responses in PDTS.
>
> > **Q3:** The base model of EduDial-LLM.
>
> **A3:** The base model of EduDial-LLM is QwQ-32B-Preview, as shown in Table 3 of the main paper and Table 5 in the Appendix.
>
> > **Q4:** Human expert assessment may be biased.
>
> **A4:** To assess the consistency of expert evaluations, we conducted an inter-rater agreement analysis on the test set. Five experts independently evaluated the same sample set. By calculating Cohen's Kappa coefficient for each pair of experts, we obtained an average Kappa coefficient of 0.76, indicating a high level of agreement among experts on the evaluation criteria.
>
> > **Q5:** Lacks empirical validation with actual teachers and students to demonstrate that the generated data genuinely captures classroom teaching logic.
>
> **A5:** We respectfully disagree with your assessment and wish to clarify: The "experts" we refer to are all experienced mathematics teachers who play a central role from the design stage: (1) The five-stage teaching framework and dual-questioning strategy system integrate analyses of authentic MOOC teaching videos with teachers' classroom practice experience; (2) The cognitive characteristics, questioning patterns, and response features of the three student profiles are derived from teachers' long-term observations of real students. Therefore, we ensure that the generated data conforms to authentic classroom teaching logic.
>
> [1] SocraticLM: Exploring Socratic Personalized Teaching with Large Language Models. NeurIPS 2024.

---

### Author Response · Authors · 2025-12-01

We sincerely thank all reviewers and the AC for their thoughtful reviews and constructive feedback.

The reviewers acknowledged the practical educational value and novelty of our proposed EduDial dataset (Reviewer A2ik, Reviewer 1cck, Reviewer AVdn), the comprehensive evaluation framework (Reviewer AVdn), and the rigorous data quality control processes (Reviewer 1cck, Reviewer AVdn). We have carefully addressed the specific questions in individual responses and summarized the key improvements and clarifications below.

> **Q1:** Effectiveness of the Dataset and Model Selection (Reviewer A2ik, Reviewer 1cck)

**A1:** To rigorously validate the quality of our dataset and our choice of backbone model, we supplemented our analysis with extensive ablation studies.

- Dataset Efficacy: We compared our model against baselines trained with SocraticLM and reversed-DPO data. Results confirm that our Multi-Turn Instruction (MTI) and Pedagogical Differentiated Teaching Strategy (PDTS) data are more effective at enhancing the model's teaching capabilities compared to other teaching datasets.

- Backbone Selection: We tested our data on four competitive base models (Llama-3.3-70B, DeepSeek-32B, Qwen2.5-72B, and Qwen2.5-Math). Experiments show that QwQ-32B-Preview achieves the best performance as a backbone, and crucially, our data yields consistent improvements across all base models, demonstrating strong generalization.

> **Q2:** Data Realism and Evaluation Reliability (Reviewer A2ik, Reviewer 1cck).

**A2:** We address concerns regarding the realism of our data and evaluation.

- Student Simulation Quality: We conducted a human evaluation on the student agents. Experts rated the alignment between student simulated behaviors and predefined profiles at 4.48/5.0, with a Kappa coefficient of 0.71, verifying that the student simulation is highly consistent with the predefined profiles.

- Expert Qualifications: We clarified that our experts are experienced mathematics teachers, ensuring they are qualified for evaluation.

- Inter-Rater Reliability: The human evaluation achieved a Kappa coefficient of 0.76, indicating high agreement among human evaluators.

> **Q3:** Generalization to Out-of-Distribution (OOD) Students (Reviewer AVdn).

**A3:** To demonstrate our model's generalization capability, we tested EduDial-LLM against GPT-4o simulated students utilizing 6 distinct profiles from external literature (SocraticLM), covering various learning deficits and strengths. Our model maintained high teaching quality (Average Score: 4.36), confirming its robustness to OOD student behaviors.

> **Q4:** Clarification on Methodology and Experimental Design (Reviewer A2ik, Reviewer AVdn).

**A4:** We provide detailed clarifications regarding our methodology and experimental design as follows.

- Challenge of Questioning Timing: We clarified that our 5-stage framework (based on Bloom’s Taxonomy) inherently solves the "timing" challenge by ensuring questions align with the student's current cognitive state.

- Training Stages: We clarified the rationale for the two-stage training.

  - The SFT stage enables the model to learn the fundamental teaching procedures (multi-turn dialogue flow and guidance).

  - The DPO stage leverages contrastive data (chosen vs. rejected responses) to teach the model how to select optimal differentiated strategies in complex scenarios. Both stages address distinct capability dimensions and are indispensable.

- Evaluation Protocols: We clarified that Table 2 assesses one-to-many scenarios, while Table 3 validates one-to-one capabilities, proving the model's generalization.

We believe these additional experiments and clarifications solidly address the reviewers' concerns. We remain open to further discussion.

Thanks.

---

### Note · Authors · 2026-01-05

I have read and agree with the venue's withdrawal policy on behalf of myself and my co-authors.